# Coronavirus Disease 2019 (COVID-19) in a Patient with Disseminated Histoplasmosis and HIV—A Case Report from Argentina and Literature Review

**DOI:** 10.3390/jof6040275

**Published:** 2020-11-10

**Authors:** Fernando A. Messina, Emmanuel Marin, Diego H. Caceres, Mercedes Romero, Roxana Depardo, Maria M. Priarone, Laura Rey, Mariana Vázquez, Paul E. Verweij, Tom M. Chiller, Gabriela Santiso

**Affiliations:** 1Unidad de Micología, Hospital de Infecciosas F. J. Muñiz B.A. Uspallata 2272, C1282 CABA, Argentina; fmessina35@gmail.com (F.A.M.); emamarin@hotmail.com (E.M.); mecharomero@gmail.com (M.R.); roxanadepardo@yahoo.com.ar (R.D.); 2Mycotic Diseases Branch, Centers for Disease Control and Prevention (CDC), Atlanta, GA 30333, USA; diegocaceres84@gmail.com (D.H.C.); tnc3@cdc.gov (T.M.C.); 3Department of Medical Microbiology, Radboud University Medical Center and Center of Expertise in Mycology Radboudumc/CWZ, 9101 Nijmegen, The Netherlands; Paul.Verweij@radboudumc.nl; 4División de HIV SIDA, Hospital de Infecciosas F. J. Muñiz B.A., Uspallata 2272, C1282 CABA, Argentina; maiapriarone@yahoo.com; 5División de Neumonología, Hospital de Infecciosas F. J. Muñiz B.A., Uspallata 2272, C1282 CABA, Argentina; lauradianarey@gmail.com; 6Bacteriología de Guardia, Hospital de Infecciosas F. J. Muñiz B.A., Uspallata 2272, C1282 CABA, Argentina; mariana.belenvazquez@gmail.com

**Keywords:** Histoplasmosis, *Histoplasma*, COVID-19, Coronavirus

## Abstract

The disease caused by the new SARS-CoV-2, known as Coronavirus disease 2019 (COVID-19), was first identified in China in December 2019 and rapidly spread around the world. Coinfections with fungal pathogens in patients with COVID-19 add challenges to patient care. We conducted a literature review on fungal coinfections in patients with COVID-19. We describe a report of a patient with disseminated histoplasmosis who was likely infected with SARS-CoV-2 and experienced COVID-19 during hospital care in Buenos Aires, Argentina. This patient presented with advanced HIV disease, a well-known factor for disseminated histoplasmosis; on the other hand, we suspected that COVID-19 was acquired during hospitalization but there is not enough evidence to support this hypothesis. Clinical correlation and the use of specific *Histoplasma* and COVID-19 rapid diagnostics assays were key to the timely diagnosis of both infections, permitting appropriate treatment and patient care.

## 1. Introduction

Coronaviruses are positive-sense RNA viruses and are named for the crown-like spikes on their surface. There are hundreds of coronaviruses, which usually infect animals, birds and various mammals, like camelids, bats, civets, rats, mice, cats and dogs [1]. The first report of human coronavirus disease was in the mid-1960s. Since this first report, there have been seven coronaviruses reported to infect humans and four of these usually cause mild to moderate symptoms of the upper respiratory tract, such as the common cold. In the past two decades, the other three coronaviruses have been recognized as causing serious disease in humans, leading to two large outbreaks and the current global pandemic [1,2,3].

The first known large outbreak of coronavirus disease occurred in 2002 in the Guangdong province, China. This outbreak of Severe Acute Respiratory Syndrome (SARS) spread to five continents, affecting 8098 people and causing a total of 774 deaths [4]. A second large outbreak of coronavirus occurred in 2012 and emerged on the Arabian Peninsula and was given the name Middle East Respiratory Syndrome (MERS-CoV). MERS-CoV cases were reported in 27 countries. Since 2012, the World Health Organization (WHO) has reported a total of 2494 infections and 858 deaths [5,6]. In December 2019, a novel coronavirus (SARS-CoV-2), which is the causative agent of coronavirus disease 2019 (COVID-19), was identified in Wuhan, China. The WHO declared this a global pandemic on 11 March 2020 [7,8]. As of the end of October 2020, the COVID-19 pandemic has spread to 216 countries, areas or territories, has affected more than 45 million people and has caused more than a million deaths worldwide [8]. In Argentina, the first case of COVID-19 was identified on 3 March 2020 and by 31 October 2020 the Argentinean Ministry of Health reported more than one million cases and 30,000 deaths [9].

Histoplasmosis, a disease caused by the dimorphic thermal fungus *Histoplasma capsulatum*, is frequently reported in the Americas but recently is increasingly reported in regions outside the “traditional” endemic region [10,11,12]. Histoplasmosis presents with a wide range of clinical manifestations, from flu-like symptoms to multiple manifestations of disseminated histoplasmosis (DH). DH principally affects individuals with immunodeficiencies, especially people living with HIV (PLHIV) with advanced disease [13].

## 2. Case Report and Results

In early June 2020, a 36-year-old female with HIV was admitted at the emergency service after experiencing 15 days of dyspnea, cough and night sweats. HIV was diagnosed ten years prior and the patient was not adherent to HIV antiretroviral therapy (ART). The last ART treatment was in June 2018 (atazanavir/ritonavir, tenofovir/emtricitabine). The patient was a heavy smoker, more than 30 cigarettes/day since she was 14 years old and also a person who uses drugs (marijuana and cocaine). Over the two months prior to hospital admission she had developed asthenia and adynamia. On admission (day 1) the patient had fever (temperature 38.2 °C), a respiratory rate of 20 breaths per minute, oxygen saturation of 95%, a heart rate of 106 beats per minute and a blood pressure of 110/80 mm Hg. She presented with hypoventilation and skin and mucosal lesions, including papule-like lesions near her right malar and nose area and an erythematous lesion slightly ulcerated in the central palate. The patient also presented with a bilateral and diffuse micronodular interstitial pattern on the chest computed tomography (CT, Figure 1). Laboratory analysis showed a low white blood cell count (3200 cells/mm^3^) and low hemoglobin and hematocrit (10.1 g/dL and 31% respectively) and a normal platelet count (191,000 cells/mm^3^). Serum biochemistry analysis was normal. Her HIV viral load was 356,000 copies/mm^3^ and a CD4 cell count of 3 cells/mm^3^ (Figure 2).

On admission at the emergency department and due the low CD4 cells count, samples for tuberculosis (TB) and mycology analysis were collected, including sputum, blood cultures (processed by lysis centrifugation), sera and urine. Specimens were tested for *Histoplasma* and *Cryptococcus* antigen detection (Clarus *Histoplasma* GM enzyme immunoassay and CrAg^®^ LFA cryptococcal antigen, both products from IMMY, Norman, OK, USA) and immunodiffusion method for detection of *Histoplasma* specific antibodies. Sputum smear microscopy was negative for acid-fast bacilli (AFB) using Ziehl-Neelsen (ZN) stain but intracellular clusters of budding yeast compatible with *H. capsulatum* were observed using Wright-Giemsa stain (Figure 1). Results of antigen testing came back within 24 h. *Cryptococcus* antigen was negative and *Histoplasma* antigen was positive, with a concentration of 24.7 ng/mL in urine and 5.10 ng/mL in serum (Figure 2 and Figure 3). Based on these laboratory findings, the diagnosis of histoplasmosis was established and the patient started treatment with deoxycholate amphotericin B (dAmB) in a dose of 0.7 mg/kg per day [14]. Following institutional protocols, a nasopharyngeal swab sample was collected and tested for SARS-CoV-2 by Real-time polymerase chain reaction (RT-PCR), which was negative and the patient had no history of contact with a person with suspected or confirmed COVID-19 (Figure 2).

After three days of dAmB (day 3 of admission), patient became afebrile, with normal heart and respiratory rate and oxygen saturation of 95%. Immunodiffusion testing was non-reactive for *Histoplasma* antibodies. Renal function parameters showed normal blood levels of creatinine, sodium and potassium and slightly elevated blood urea nitrogen. After 7 days of dAmB therapy, mucosal lesions disappeared and the skin lesions improved and antifungal therapy was switched to itraconazole capsule (200 mg twice daily), as is indicated on the international histoplasmosis guidelines [14]. On the same day, the patient developed a new fever (38.6 °C), without alteration of respiratory rate and oxygen saturation of 97% and without other symptoms. A diagnostic workup was performed including blood cultures and a second nasopharyngeal swab sample for SARS-CoV-2 testing. The SARS-CoV-2 RT-PCR test was positive and the patient was moved to a COVID-19 ward. Potential risk factors for severe COVID-19 were not identified in this patient and the disease presented with mild symptoms, so only monitoring of symptoms was done on this patient (Figure 2).

On day 12, the patient had been afebrile for four days prior, with normal heart and respiratory rate and oxygen saturation of 97%. Blood cultures remained negative. On day 14, the patient continued to be afebrile and follow-up *Histoplasma* antigen testing in urine and serum showed decreasing antigen levels (Figure 2 and Figure 3). Further decrease of *Histoplasma* antigen was noted on day 21, indicating a good response to antifungal therapy (Figure 3) and skin lesions disappear. On day 23, which was 15 days without fever and presenting resolution of symptoms, the patient was discharged with itraconazole capsule 400 mg/day and started ART, following the Hospital de Infecciosas F. J. Muñiz’s protocol for ART initiation, with dolutegravir, emtricitabine and tenofovir. Twenty-seven days after hospital discharge (day 51), the patient attended a follow-up consultation. Physical evaluation showed further clinical improvement of the patient’s medical condition and *Histoplasma* antigen testing in urine and sera showed a further decrease of antigen levels (10.6 ng/mL and 1.22 ng/mL, respectively).

## 3. Fungal Infections and COVID-19: Literature Review

A literature search on PubMed Central was done on 31 July 2020. We searched for the following terms—COVID-19; Coronavirus, SARS-CoV-2, fungal infections, mycoses, histoplasmosis, *Histoplasma*, aspergillosis, *Aspergillus*, candidiasis, *Candida, Pneumocystis*, coccidioidomycosis and *Coccidioides* and included studies published in English, Spanish and Portuguese. We also did a forward and backward “snowballing” reference chain search. We searched for case reports of cohorts of patients with COVID-19 with co-infection with fungal diseases.

We identified a total of 28 references describing COVID-19 cases co-infected with fungal diseases, including 16 case series and 12 case reports [15,16,17,18,19,20,21,22,23,24,25,26,27,28,29,30,31,32,33,34,35,36,37,38,39,40,41,42]. These reports came from 15 countries; half were from European countries (54%), followed by reports from Asia (28%), the Americas (14%) and Oceania (4%); no reports from Africa were identified. By diagnosis, 17 (61%) reports described infections by molds, 16 caused by *Aspergillus* species and one caused by *Fusarium proliferatum*, three reports described infections caused by yeasts (*Candida* species and *Saccharomyces cerevisiae*), two reports of *Pneumocystis* pneumonia and one report of coccidioidomycosis. Five reports described fungal infections in a hospital series of COVID-19 cases, mostly including infections caused by *Aspergillus* and *Candida* species but also cases of mucormycosis and cryptococcosis (Table 1). COVID-19 associated pulmonary aspergillosis (CAPA) was described in 21 of the 28 references reviewed [15,16,17,18,19,20,21,22,23,24,25,26,27,28,29,30,38,39,40,41,42]. In summary, a total of 164 cases of CAPA were described, most were caused by *A. fumigatus* and a few by other *Aspergillus* species (*A. flavus* and *A. penicillioides*). CAPA case series reported prevalence ranging from 8% to 39% and mortality rates ranging from 44% to 67%. All CAPA cases were reported in severe COVID-19 patients (Table 1). Only one report described a SARS-CoV-2 infection in a PLHIV, who developed *Pneumocystis* pneumonia, based on chest CT and elevated lactate dehydrogenase (Table 1) [36]. Only one report described a co-infection with a dimorphic fungus, *Coccidioides*. This was a report of a patient who did not adhere to chronic coccidioidomycosis treatment, who developed moderate COVID-19, no hospitalization was needed in this case [37].

## 4. Discussion

COVID-19 associated with fungal diseases has been reported. These reports mostly describe cases of CAPA and to a much lesser extent other fungal pathogens [15,16,17,18,19,20,21,22,23,24,25,26,27,28,29,30,31,32,34,35,36,37,38,39,40,41,42]. The majority of cases are opportunistic secondary infections to COVID-19. We report a case of endemic mycosis in a patient who was likely infected with SARS-CoV-2 while in the hospital. The treatment of histoplasmosis was successful despite the patient having a very poor immune status due to advanced HIV disease and SARS-CoV-2 infection. This case highlights that the diagnosis of fungal infections is even more challenging during the current COVID-19 pandemic, where strategies focusing on containing the pandemic, have significantly reduced access to non-COVID-19–related health care services [43,44,45]. In addition, medical procedures such as bronchoscopy and bronchoalveolar lavage (BAL) are restricted due to generation of aerosols. These procedures are critical to diagnose other or secondary pulmonary infections, including invasive fungal diseases [43,44,45,46,47,48].

Among other causes of morbidity and mortality in patients with COVID-19, the frequency and impact of fungal co-infections is still limited, although the number of clinical cases and reports are increasing. It is important to note that mortality reported across these studies is concerning. It is still premature to link these fungal co-infections as significant causes of deaths but it should be noted that the risk factors related to mechanical ventilation, prolonged use of corticosteroids and lung damage caused by COVID-19 together with the cytokine storm could set up these patients for invasive fungal diseases [38,39,43,49,50,51]. Different than the typical invasive fungal infections (IFI) where COVID-19 may be a host risk factor, this reported case presented with disseminated histoplasmosis and then experienced COVID-19; this secondary viral infection did not affect histoplasmosis treatment and clinical evolution. Strict adherence to hospital mitigation strategies is needed to help reduce risk for SARS-CoV-2 transmission to hospitalized patients at higher risk for severe COVID-19. The immunological profile in patients with COVID-19 has been extensively described. Well known are the systemic inflammatory response, exacerbation of serum inflammatory bio-markers, such as C-reactive protein (CRP), lactic dehydrogenase (LDH), ferritin, D-dimer and IL-2R, IL-6, IL-10, TNF-alpha [52,53]. For PLHIV with COVID-19, information about disease severity and patient outcomes is contradictory and several reports have shown that the mortality and severity of COVID-19 in PLHIV were not increased while other studies report worse evolution in patients with both infections. It is important to mention that most of these studies have reported several methodological limitations, especially the non-evaluation of residual confounders [54,55,56,57,58]. In PLHIV it is important to consider the presence of pulmonary fungal coinfections, especially due to the similarity in chest tomographic patterns. In the literature review, we identified a report of *Pneumocystis* pneumonia (PjP) in a patient newly diagnosed with advanced HIV disease and with severe COVID-19. As important findings on this report, the patient presented reticular changes on the chest radiology with alteration of lactate dehydrogenase, a biomarker usually altered with PjP. These findings added the clinical suspicion of PjP and addressing additional laboratory testing for the confirmation of this coinfection [36]. But, it is also important to mention that similar situations have been reported in persons who are not living with HIV [35,37].

Based on laboratory and epidemiological evidence, we hypothesize that SARS-CoV-2 infection in the patient that we describe in this report happened during hospitalization. But it is important to note that the sensitivity of SARS-CoV-2 RT-PCR in nasopharyngeal swab samples was reported around 70% and it is therefore unknown whether the first negative result was not a false-negative result [59,60,61,62]. There is high risk of COVID-19 spread in hospital, so CDC is constantly updating COVID-19 infection control recommendation. Current recommendations for strengthening standard hospital infection control practices are aimed to reduce nosocomial spread of COVID-19 and protect health care workers [63]. In this case it is particularly important to highlight the role of rapid and accurate diagnostic assays in diagnosis and care of histoplasmosis and COVID-19. Additionally, in this case, *Histoplasma* antigen testing was crucial in several aspects: (1) Analysis of urine and serum samples represent lower risk of COVID-19 exposure of medical staff to COVID-19; (2) these types of samples are less invasive than other types of samples traditionally used to diagnose histoplasmosis, like BAL or bone marrow aspirate; (3) antigen test was useful for evaluation of the response of the *Histoplasma* specific antifungal treatment. All these findings correlate with the published guidelines for histoplasmosis diagnosis and treatment in PLHIV [14,64].

## 5. Conclusions

Here we describe a case of disseminated histoplasmosis and COVID-19 in a person with advanced HIV disease. It is well known that advanced HIV is one of the main risks factors for the development of disseminated histoplasmosis. In this patient we suspected that SARS-CoV-2 infection happened during hospitalization but based on information available, it is not possible to proof this hypothesis. Based on this experience, we summarized the main findings of this case and additional evidence published in the literature, aimed to inform other clinicians about fungal co-infection on patients with COVID-19. We highlight the importance of patient’s epidemiological and clinical evaluation, supported by rapid and highly accurate diagnostic assays.

## Figures and Tables

**Figure 1 jof-06-00275-f001:**
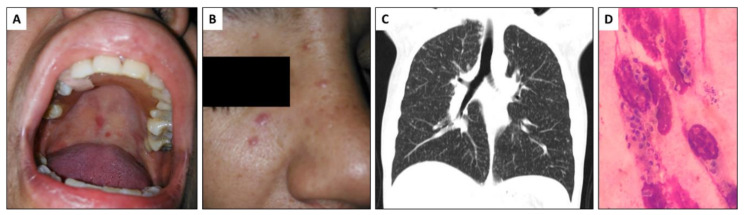
Clinical findings: (**A**,**B**) Papule-like lesions in the right malar region and in the proximal area of the nose and erythematous lesion with slight central ulceration in the central palate. (**C**) Chest computed tomography (CT), without contrast, revealed a bilateral and diffuse micronodular interstitial pattern, compatible with miliary histoplasmosis. (**D**) Wright-Giemsa stain on sputum smear, intracellular clusters of budding yeast compatible with *H. capsulatum* using Wright-Giemsa stain.

**Figure 2 jof-06-00275-f002:**
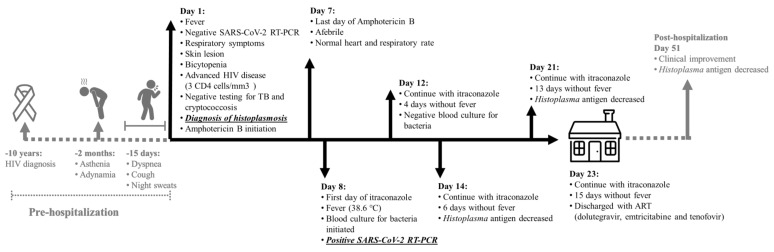
Case timeline: Coronavirus disease 2019 (COVID-19) in a patient with progressive disseminated histoplasmosis from Argentina.

**Figure 3 jof-06-00275-f003:**
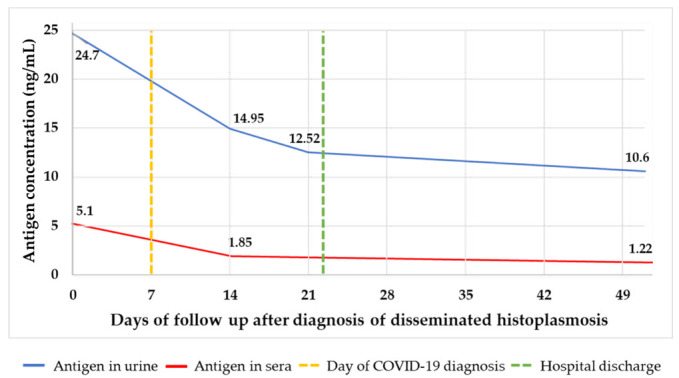
Follow-up using *Histoplasma* antigen testing in a patient with disseminated histoplasmosis and COVID-19.

**Table 1 jof-06-00275-t001:** Results of the literature review on fungal diseases and COVID-19.

Ref	Country	Report Summary
**Molds (*Aspergillus*/*Fusarium*)**		
[15]	Italy *	108 patients with severe COVID-19, 30 case-patients with (28% prevalence). Dx+%: 100% GM in BAL (pooled positivity), 67% *Aspergillus* PCR, 63% culture and 3% GM in serum. 30-day mortality was 44%.
[16]	France *	Evaluation of molecular testing for detection of aspergillosis in ICU patients. Using the AspICU algorithm, a total of nine patients were classified as PIPA (20% prevalence). Adding PCR results, a total of 15 patients were classified as PIPA (33% prevalence).
[17]	France *	Nine CAPA cases on 29 severe COVID-19 (33% prevalence). Dx+%: 78% culture, 44% fungal PCR, 29% GM in BAL, 11% GM in serum. 44% mortality.
[18]	Pakistan *	Nine aspergillosis cases on 23 severe COVID-19 patients (39% prevalence). Five were defined as CAPA and four were defined as colonized. GM was negative in all patients; BG was positive only in one patient. Three of the five patients with CAPA die (60%).
[19]	China *	Eight aspergillosis cases on 104 patients with COVID-19 (8% prevalence). All cases diagnosed by culture. Not mortality reported.
[20]	Belgium *	Seven CAPA cases on 20 COVID-19 patients on mechanical ventilation (35% prevalence). Dx+%: 86% culture, 100% GM in BAL, 0% GM in serum. 57% mortality.
[21]	Netherlands *	Six CAPA cases on 31 severe COVID-19 patients (19% prevalence). Dx+%: 83% culture, 50% GM in BAL, 0% GM in serum. 67% mortality.
[22]	Germany *	Five CAPA cases on 19 patients with severe COVID-19 (26% prevalence). Dx+%: 80% fungal PCR, 60% GM in BAL, 60% culture, 40% GM in serum. 60% mortality.
[23]	France ^ф^	Case report on a critically ill COVID-19 patient. Diagnosis was done by positive PCR on TA; GM was also positive in serum and TA. BG was negative. *A. fumigatus* (identified by MALDI-TOF) was isolated on second TA. Patient died.
[24]	France ^ф^	Description of the first five cases of COVID-19 in France. One CAPA patient, *A. flavus* isolated from TA culture. Patient initially was treated with voriconazole and then switched to isavuconazole. Patient also have co-infected with *A. baumannii*. Patient died.
[25]	Italy ^ф^	Case report of IPA. Patient diagnosed by positive culture on BAL (*A. fumigatus*) and positive serum GM (index: 8.6). Post-mortem lung examination confirmed IPA.
[26]	Netherlands ^ф^	CAPA report of an azole-resistant *Aspergillus fumigatus* isolated from TA. TA was also GM positive (>3.0 index). In serum, BG was positive (1590 pg/mL) and GM negative. Patient died.
[27]	Austria ^ф^	IPA in a critically ill COVID-19 patient. *A. fumigatus* isolated from TA and positive antigen using *Aspergillus* Ag LFA. Serum GM and BG were negative. Patient died after three days of antifungal treatment.
[28]	Argentina ^ф^	Case report of ventilator-associated pneumonia involving *A. flavus*. Patient diagnosed by culture and positive serum GM. Multiple co-infections (*E. faecalis, A. baumanii*, coagulase-negative staphylococci and *C. lusitaniae*). Patient died.
[29]	Brazil ^ф^	Case report of postmortem IPA caused by *Aspergillus penicillioides*. Diagnosis was done by histopathological analysis of lung tissue and PCR with sequencing (ITS1 and ITS2 gen targets).
[30]	Australia ^ф^	COVID-19 associated pulmonary aspergillosis caused by *A. fumigatus* isolated from non-bronchoscopy endotracheal aspirate. Patient also presented positive blood culture for *Facklamia hominis* and urine culture for *Escherichia coli*. Patient survived.
[31]	France ^ф^	ICU patient with acute respiratory failure. After seven days intubated, a mold grew (7 × 102/CFU) on BAL (no bacteria were detected). MALDI-TOF identified mold as *Fusarium proliferatum*. Patient got positive GM (index: 1.7). After 30 days patient was extubated and translated to a post-ICU area.
**Yeast (*Candida*/*Saccharomyces*)**		
[32]	Iran *	53 (5%) out of 1059 patients with confirmed COVID-19 infection had OPC. *C. albicans* was the most isolated species (71%), followed by *C. glabrata* (11%), *C. dubliniensis* (9%), *C. parapsilosis* (5%), *C. tropicalis* (3%) and *Pichia kudriavzevii* (2%). Most isolates were susceptible to all three classes of antifungal drugs.
[33]	India *	Fifteen (2.5%) BSI among 596 ICU patients. *C. auris* caused 10 (67%) BSI, followed by *C. albicans* (*n* = 3), *C. tropicalis* (*n* = 1) and *C. krusei* (*n* = 1). Eight of 15 (53%) BSI cases died. Six of 10 (60%) of patients with *C. auris* deceased.
[34]	Greece *	Two ICU patients with bloodstream infection caused by *Saccharomyces*. Cases were linked with diarrhea treatment using Ultra-Levure (preparation of *S. cerevisiae*). Patients survived.
**Opportunistic (*Pneumocystis*)**		
[35]	USA ^ф^	Case report of *Pneumocystis* pneumonia (PcP) and COVID-19. Patient reported two weeks of respiratory symptoms at hospital admission, non-HIV infected, with low CD4 T cells count (291 cells/µL). Patient had elevated BG (305 pg/mL) and a positive specific *Pneumocystis* real-time PCR. After one week of trimethoprim-sulfamethoxazole treatment BG was significantly reduced (90 pg/mL). After 10 days CD4 T cells improved (730 cells/µL).
[36]	Germany ^ф^	Case report of *Pneumocystis* pneumonia in a newly diagnosed HIV patient who developed acute respiratory failure due the COVID-19 infection. On this patient the presence of fine reticular changes in the chest computed tomography and elevated lactate dehydrogenase added the clinical suspicion of *Pneumocystis* pneumonia.
**Dimorphic (*Coccidioides*)**		
[37]	USA ^ф^	Case report of a chronic cavitary pulmonary coccidioidomycosis nonadherent to therapy and medical controls. Patient attended primary provider after having 6 days of fever, cough and body aches. First was diagnosed with bronchitis and the provider prescribed azithromycin. Two days later patient went to the emergency for weakness, progressive cough, fever and body aches. Laboratory testing confirmed COVID-19; patient also presented elevated specific *Coccidioides* Ab. Hospitalization was not required.
**Various Infections**		
[38]	UK *	836 patients with COVID-19 (February to April 2020). *Candida* spp. and other yeast were isolated from respiratory samples on 24 (21%) of 112 patients tested. Three patients developed *C. albicans* fungemia (all central line–associated infections). Three patients with *A. fumigatus* culture were identified; one patient was classified as colonized, the other two were classified as possible infection (both were negative GM and BG in serum).
[39]	China *	257 COVID-19 cases (January to February 2020). On 243 (94%) of patients were identified co-infections by other virus, bacteria or fungal pathogens. Fungal pathogens detected included: *Aspergillus* (60, 23%), *Mucor* (6, 2%), *Candida* (2, 0.8%) and *Cryptococcus* (1, 0.4%).
[40]	China *	99 COVID-19 cases (January 2020). Four (4%) patients were diagnosed with fungal co-infections (aspergillosis and candidiasis). 15 (15%) patients were under antifungal treatment.
[41]	China *	52 COVID-19 critically ill patients. In two patients, *A. flavus* and *A. fumigatus* were isolated from respiratory tract secretions (found in one patient each). In one patient *C. albicans* was isolated from urine.
[42]	China *	Description of 85 fatal cases of COVID-19. Fungal culture was positive in three of nine samples tested (33%), it was not described fungus isolated. 13% of cases were treated with antifungals.

Abbreviations: (Ref) Reference; (*) case series; (^ф^) case report; (CAPA) COVID-19 associated pulmonary aspergillosis; (Dx+%) Diagnostics positivity rate: (GM) Galactomannan; (BAL) Bronchoalveolar lavage; (PCR) Polymerase chain reaction; (AspICU) Clinical algorithm to diagnose invasive pulmonary aspergillosis in critically ill patients; (PIPA) Putative invasive pulmonary aspergillosis; (IPA) Invasive pulmonary aspergillosis; (OPC) Oropharyngeal candidiasis; (BSI) bloodstream infections; (BG) (1→3)-β-d-Glucan; (TA) Tracheal aspirate; (Ag) antigen; (LFA) Lateral flow assay; (UK) United Kingdom; (USA) United States of America.

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
