# Peer review of "Coronavirus Disease 2019 (COVID-19) in a Patient with Disseminated Histoplasmosis and HIV—A Case Report from Argentina and Literature Review"

_jof, 2020, doi:10.3390/jof6040275_

Round 1

Reviewer 1 Report

Links between covid19 and fungal infections appear to deserve more attention than they have received so this study has value. However I am concerned that the index case has very severe HIV disease and the articles reviewed barely mention HIV.

In the index patient, the sequence was HIV-> fungal infection-> covid

So are you suggesting that fungal infections are a risk factor for covid

or

HIV is a risk factor for covid

or

hospitalisation is a risk factor for covid?

It may be impossible to know...but the issue should be addressed systematically as it is more interesting than a conclusion that "rapid tests" are valuable (we all know that). I suggest you start by shortening the manuscript and adding informative sub-headings to chart the arguments  about which comes first.

The case report can be much briefer but should explain the rationale for the delayed start of ART....and the choice of ART drugs.

Author Response

Reviewer: 1

1. Links between covid19 and fungal infections appear to deserve more attention than they have received so this study has value. However I am concerned that the index case has very severe HIV disease and the articles reviewed barely mention HIV.

In the index patient, the sequence was HIV-> fungal infection-> covid

So are you suggesting that fungal infections are a risk factor for covid

or

HIV is a risk factor for covid

or

hospitalisation is a risk factor for covid?

It may be impossible to know...but the issue should be addressed systematically as it is more interesting than a conclusion that "rapid tests" are valuable (we all know that). I suggest you start by shortening the manuscript and adding informative sub-headings to chart the arguments about which comes first.

  1. Thanks for your comments. We can affirm that the advanced HIV was the risk factor for histoplasmosis, but it is hard to make the same conclusion with COVID-19. We suspect that this infection was acquired during hospitalization, because this patient presented a negative SARS-CoV-2 PCR at admission and did not reported any epidemiological link with COVID-19. But, in the other hand, we know that SARS-CoV-2 PCR could present false negative results. We added the following statements in manuscript abstract and conclusion:
  • Abstract: “This patient was presented with advanced HIV disease, a well know factor for disseminated histoplasmosis, on the other hand, we suspected that COVID-19 was acquired during hospitalization, but there is not enough evidence to support this hypothesis.”
  • Conclusion: “It is well known that advanced HIV is one of the main risks factors for the development of disseminated histoplasmosis. In this patient we suspected that SARS-CoV-2 infection happened during hospitalization, but based on information available, it is not possible to proof this hypothesis.”

About your comment of shortening the manuscript, we would like to keep it in the current format.

  1. The case report can be much briefer but should explain the rationale for the delayed start of ART....and the choice of ART drugs.
    R. Instauration of ART treatment was done following Hospital de Infecciosas F. J. Muñiz protocol. This protocol request before start ART a psychological evaluation of patients, in special drug user. This evaluation is done in order to evaluated patient adherence to HIV treatment. ART drugs are selected bases on results of this is psychological evaluation and drug availability in the hospital.

We modified ART statement as follows: “… and started ART, following Hospital de Infecciosas F. J. Muñiz protocol for ART initiation, with dolutegravir, emtricitabine and tenofovir...”    

Reviewer 2 Report

There are both minor and major issues with this manuscript

  1. Since this is meant to be a review, the authors should include other animals in which SARS-Cov-2 has been isolated in addition to those listed in the one reference cited. The CDC has reported infection in minks, lions and other animals; I recommend a literature search to delineate all the animals in which disease has been identified since it is mentioned.
  2. I think that it is standard for documentation that patients, when they have very low CD4 counts and certain opportunistic infections that they are classified as having AIDS. Nowhere does this author indicate this despite the CD4 count of only three cells/uL and disseminated histoplasmosis, both AIDS defining laboratory values/diseases.  I do not think the euphemism “advanced disease“ is appropriate in this circumstance.
  3. Line 60 the intent is to identify illicit or recreational drug use, and this should specifically be stated.
  4. 4 first paragraph, the authors need to explicitly state from the beginning what specimens are being tested for which analytes Grouping the histo and crypto testing together when they are probably tested on different specimens makes this especially confusion.  This needs to be cleaned up.
  5. What is the utility of testing both urine and serum for histoplasma antigen? It is clear that based on urine testing alone, the counts are higher longer and one has a better chance of make the diagnosis compared to testing on serum.  It would be worthwhile to point this out.  This is an egregious waste of money and puts the phlebotomists at risk (especially when the viral load is so high) and contributes nothing additional to the case that blood samples were repeatedly collected and analyzed for this patient.
  6. The authors need to make the case that these fungal infections are because of the patient’s COVID-19 infections. To do that, we need to know that there is not some other underlying immunocompromising risk factor to account for the increase risk for acquiring invasive fungal infections.  Do these patients from the literature have diabetes?  Autoimmune diseases?  Cancer? Transplantation? Immune-modulating medications?  HIV/AIDS?  Leukemia/Lymphoma?  Are they elderly?  Clearly, the patient’s underlying risk factor for having Histo for the case presented is the diagnosis of AIDS due to the very low CD4 count.  The authors appear to be suggesting that the cause for the invasive fungal infections in the setting of COVID-189 is the viral infection rather than some other underlying immunocompromising condition.   Table 1 does not provide enough information to determine what the underlying illnesses are for the severe disease, although I am sure the authors do go into this as part of their manuscripts.  I would be very surprised if there were not additional immunocompromising situations for these patients with invasive fungal infections. 

Author Response

Reviewer: 2

1. Since this is meant to be a review, the authors should include other animals in which SARS-Cov-2 has been isolated in addition to those listed in the one reference cited. The CDC has reported infection in minks, lions and other animals; I recommend a literature search to delineate all the animals in which disease has been identified since it is mentioned.

R. Thanks for your comment, but we consider that a systematic review is animals it is not part of the goal of this case report.    

2. I think that it is standard for documentation that patients, when they have very low CD4 counts and certain opportunistic infections that they are classified as having AIDS. Nowhere does this author indicate this despite the CD4 count of only three cells/uL and disseminated histoplasmosis, both AIDS defining laboratory values/diseases. I do not think the euphemism “advanced disease“ is appropriate in this circumstance.

R. In 2017, the World Health Organization (WHO) published guidelines for the management of advanced human immunodeficiency virus (HIV). Advanced HIV disease is defined by the World Health Organization (WHO) as having a CD4 cell count <200 cells/µL or stage III or IV disease. We would like to keep this term following WHO recommendations.

3. Line 60 the intent is to identify illicit or recreational drug use, and this should specifically be stated.

R. Thanks for you comment, we identified drugs in page 2, line 64.

4. first paragraph, the authors need to explicitly state from the beginning what specimens are being tested for which analytes Grouping the histo and crypto testing together when they are probably tested on different specimens makes this especially confusion. This needs to be cleaned up.

R. Thanks, we modified the statement as follows: “On admission at the emergency department, and due the low CD4 cells count, samples for tuberculosis (TB) and mycology analysis were collected, including sputum, blood cultures

5. What is the utility of testing both urine and serum for histoplasma antigen? It is clear that based on urine testing alone, the counts are higher longer and one has a better chance of make the diagnosis compared to testing on serum. It would be worthwhile to point this out. This is an egregious waste of money and puts the phlebotomists at risk (especially when the viral load is so high) and contributes nothing additional to the case that blood samples were repeatedly collected and analyzed for this patient.

R. The comparison of Histoplasma antigen in serum is part of the new knowledge of this report. We aimed to compared Histoplasma antigen concentration in urine and serum. And with regard to risk, phlebotomists follow all hospital protocols for biosecurity. Addtionally, it is important to said, that blood extraction are frequently done during hospitalization, in special in patients receiving amphotericin B, as part pf the evaluation of the kidney and liver function.

6. The authors need to make the case that these fungal infections are because of the patient’s COVID-19 infections. To do that, we need to know that there is not some other underlying immunocompromising risk factor to account for the increase risk for acquiring invasive fungal infections. Do these patients from the literature have diabetes? Autoimmune diseases? Cancer? Transplantation? Immune-modulating medications? HIV/AIDS? Leukemia/Lymphoma? Are they elderly? Clearly, the patient’s underlying risk factor for having Histo for the case presented is the diagnosis of AIDS due to the very low CD4 count. The authors appear to be suggesting that the cause for the invasive fungal infections in the setting of COVID-189 is the viral infection rather than some other underlying immunocompromising condition.   Table 1 does not provide enough information to determine what the underlying illnesses are for the severe disease, although I am sure the authors do go into this as part of their manuscripts. I would be very surprised if there were not additional immunocompromising situations for these patients with invasive fungal infections.

R. Thanks for your observation. It is still early to determinate risk factors for COVID-19 and fungal infections, but in most of reports common findings included mechanical ventilation, prolonged use of corticosteroids, and lung damage caused by COVID-19. These findings are described in the discuss section page 5, line 165.

Reviewer 3 Report

In this review, the authors conducted a detailed literature review on fungal diseases in COVID-19 patients. They summarized their review with an interesting table grouping together cases of fungal and SARS-CoV-2 coinfections. The authors also described the case of a patient with disseminated histoplasmosis who was infected with SARS-CoV-2 probably during his hospitalization. They concluded that clinical correlation and the use of rapid diagnostic tests allowing appropriate treatment and patient care.

Comments and suggestions for Authors :

  • The authors mentioned that the patient had mild symptoms related to COVID-19 that only required monitoring, what were these symptoms (other than fever)?
  • The authors indicated (page 4, line 24) that potential risk factors for severe COVID-19 were not identified in this patient? This sentence may lead to confusion regarding pre-existing conditions with increased risk of severe COVID-19 already mentioned in the patient's clinical history (HIV and heavy smoking).
  • Direct cytological examination is a good laboratory tool for the rapid diagnosis of Histoplasma (sputum smear in this case report). This diagnosis can be confirmed later by antigen detection. Does your laboratory use any other special stain than the wright-Giemsa stain for the detection of yeasts ? If so which ones, please add them in the manuscript.
  • How did the mucocutaneous lesions evolve during hospitalization and treatment? or did you notice new skin lesions concomitant with SARS CoV-2 infection?

Author Response

Reviewer: 3

In this review, the authors conducted a detailed literature review on fungal diseases in COVID-19 patients. They summarized their review with an interesting table grouping together cases of fungal and SARS-CoV-2 coinfections. The authors also described the case of a patient with disseminated histoplasmosis who was infected with SARS-CoV-2 probably during his hospitalization. They concluded that clinical correlation and the use of rapid diagnostic tests allowing appropriate treatment and patient care.

Comments and suggestions for Authors :

1. The authors mentioned that the patient had mild symptoms related to COVID-19 that only required monitoring, what were these symptoms (other than fever)?

R. No other symptoms, we modified the statement as follows: “On the same day the patient developed a new fever (38.6°C), without alteration of respiratory rate and oxygen saturation of 97% and without other symptoms.”

2. The authors indicated (page 4, line 24) that potential risk factors for severe COVID-19 were not identified in this patient? This sentence may lead to confusion regarding pre-existing conditions with increased risk of severe COVID-19 already mentioned in the patient's clinical history (HIV and heavy smoking).

R. At moment of patient hospitalization, international guidelines described at tentative risk factors for COVID-19 diabetes, obesity, hypertension and age over 70 years. This patient did not meet any of these criteria.

3. Direct cytological examination is a good laboratory tool for the rapid diagnosis of Histoplasma (sputum smear in this case report). This diagnosis can be confirmed later by antigen detection. Does your laboratory use any other special stain than the wright-Giemsa stain for the detection of yeasts ? If so which ones, please add them in the manuscript.

R. The laboratory also performed Grocott's methenamine silver stain, but this stain is done in BAL samples. Wright-Giemsa, it a good stain, and also is more specific for histoplasmosis diagnosis.

4. How did the mucocutaneous lesions evolve during hospitalization and treatment? or did you notice new skin lesions concomitant with SARS CoV-2 infection?

On day seven the mucosal lesions disappeared, and the skin lesions improved. On the 21st the skin lesions disappeared. No new lesions were observed concomitant with SARS CoV-2 infection. We added this descript in the manuscript.

  • Page 4, line 102: “After 7 days of dAmB therapy, mucosal lesions disappeared and the skin lesions improved, and antifungal therapy was switched to itraconazole capsule (200 mg twice daily)”.
  • Page 4, line 102: “Further decrease of Histoplasma antigen was noted on day 21, indicating a good response to antifungal therapy (Figure 3) and skin lesions disappear.”

Round 2

Reviewer 1 Report

I would prefer the manuscript if it were more concise but this isnt raised by the other reviewers so I am happy to accept it